# Research on the Effects of Environmental Factors on the Emission of Volatile Organic Compounds from Plastic Track

**DOI:** 10.3390/ijerph20031828

**Published:** 2023-01-19

**Authors:** Gan Liu, Weitao Zheng, Hong Wang, Lin Liu, Yanrong Meng, Yu Huang, Yong Ma

**Affiliations:** 1Research Center of Sports Equipment Engineering Technology of Hubei Province, Wuhan Sports University, Wuhan 430079, China; 2Key Laboratory of Sports Engineering of General Administration of Sport of China, Wuhan Sports University, Wuhan 430079, China; 3School of Naval Architecture, Ocean and Energy Power Engineering, Wuhan University of Technology, Wuhan 430070, China

**Keywords:** sporting environment, plastic track, volatile organic compounds (VOCs), emission characteristics

## Abstract

The volatile organic compounds (VOCs) released from a plastic track can cause stimulation and damage to the human body; the temperature, relative humidity (RH) and air exchange rate (AER) have a significant impact on the release of VOCs from materials. In this study, we used a 0.1 m^3^ environmental chamber; a qualitative and quantitative analysis of VOCs released from a plastic track was conducted by gas chromatography-mass spectrometry with a temperature range of 23–60 °C, RH of 5–65% and AER of 0.5–1.5 h^−1^. The formation rate, the speciation, the nature of the main compounds and the mass concentration of VOCs under different environmental conditions were determined. It is shown that with the increase of temperature, the concentration of some main VOCs gradually increased and the *C*_alkane_ and *C*_oxygenated organic compounds_ were larger by 736.13 μg·m^−3^ and 984.22 μg·m^−3^ at 60 °C, respectively. Additionally, with the increase of RH, the concentration of different VOCs gradually increased. Nonetheless, the change in RH had no effect on the concentration percentage of different VOCs in the total VOC. With the increase in AER, the concentration of different main VOCs significantly declined, as did the VOC detection rate. When the AER was increased from 0.5 h^−1^ to 1.5 h^−1^, the *C*_alkane_ decreased by 206.74–254.21 μg·m^−3^ and *C*_oxygenated organic compounds_ decreased by 73.06–241.82 μg·m^−3^, and the number of non-detected VOC monomers increased from 1 to 7–12 species. The conclusion is that the increase in temperature and RH can promote the emission of VOCs from a plastic track, while increasing AER significantly reduces the concentrations of VOCs. Environmental temperature mainly causes the changes in the concentrations of different VOCs, and RH is a main factor leading to the variation in the detection rate of main VOCs. Overall, the release of VOCs from a plastic track is affected by environmental temperature, AER and RH in sequence. Through this paper, we clarify the effects of ambient temperature, RH and AER on the emission of VOCs from a plastic track, and furthermore, we determine the release characteristics of plastic track VOCs.

## 1. Introduction

Plastic track is widely used in various sport plants due to its good vertical deformation, impact absorption, track durability and aesthetic performance. It is mainly composed of multiple organic hydrocarbons and their derivatives, including polyurethane (PU) adhesive, rubber particles, diluents and their auxiliaries. However, the variety of the raw materials makes volatile organic compounds (VOCs) more prone to be released from the plastic track [1,2,3,4,5,6,7,8,9,10]. When people exercise on a plastic track, the released VOCs can be absorbed via breathing and skin contact, thus causing stimulation and damage to the human body, including the skin, eyes and respiratory tract, etc. [11,12,13,14].

Current research on the VOCs released from a plastic track has mainly focused on the establishment of emission detection methods [4,5,6,7,8,9,10], principal component analysis [2] and emission concentration determination [3], but the impact of environmental factors including temperature, relative humidity (RH) and air exchange rate (AER) has not been addressed in much detail.

Meanwhile, relevant studies have found a significant impact of temperature, RH and AER on the release of VOCs from materials.

As far as the role of temperature is concerned, Wal et al. [15] found a clear trend of increasing initial emission rate and decay rate of paints with increasing temperature, but the total emissions is generally consistent; this means that the baking of paints before use could accelerate the emission of pollutants. An et al. [16] used a VOC analyzer to measure their release from a wood-based panel after 96 h at 20 °C, 30 °C and 50 °C, showing that the higher temperature, the higher the emission concentration. Deng et al. [17] developed a mathematical model considering the diffusion coefficient and the temperature of porous building materials, and pointed out that the diffusion and partition coefficients of VOCs may be strongly influenced by temperature. Wei et al. [18] studied the effects of temperature (10 °C, 20 °C and 30 °C) on emission parameters (diffusion coefficient of VOCs/formaldehyde in the material, material/gas partition coefficient), indicating exponential relationships between the emission parameters and the temperature. The above studies showed that temperature can significantly change the nature of VOCs, their diffusion coefficient and their emission rate.

Concerning the role of relative humidity, Wolkoff [19], in an experiment investigating the effect of environmental factors on several VOCs emitted from five building materials, found that an increase in both temperature and RH could increase the emission of VOCs. Lin et al. [20] studied the effect of temperature and RH on the emission of VOCs from wood flooring and found that when the temperature increased from 15 to 30 °C, the VOC concentration increased 1.5–129 times. When the RH increased from 50% to 80%, the VOC concentration increased 1–32 times. Li et al. [21] studied interior latex paints and paint-based building wet materials, observing that when the RH rises from 30% to 80%, this leads many organic compounds within the wet material to hydrolyze, resulting in more pollutant organic small molecules, and the total VOC concentration increases with the increase in RH. Cao et al. [22] determined the emission characteristics and convective mass transfer coefficients of particleboard VOCs and showed that when the RH increased, the initial mass concentration of emitted VOCs increased; the convective mass transfer coefficients of the VOCs in the air also increased significantly, resulting in an enhanced dispersion of VOCs in the air. The above studies showed that humidity can exert an impact on the separation coefficient and adsorption characteristics of released VOCs.

Regarding the role of the air exchange rate, Yu et al. [23] concluded, from a theoretical mathematical model, that increasing AER would increase the release rate of pollutants, but AER would have a minimal effect. Deng et al. [24] quantitatively analyzed the effect of air exchange rate on VOCs sorption, concluding that AER is a key factor for determining whether sorption effect in building materials should be considered. Rackes et al. [25], by modeling the release of VOCs, noted that the greater the AER, the greater the release rate of VOCs from the building. Manoukian et al. [26], using statistical results from a classical two-level full factorial design, highlighted the predominant effect of AER on VOC emission factors, showing that the higher the air exchange rate (from 0.25 h^−1^ to 1.5 h^−1^), the higher the emission factor. The above studies showed that AER has a significant impact on the gas concentration gradient in the material/air boundary layer, diffusion coefficient and adsorption effect of released VOCs.

The above studies show that the temperature, RH and AER are the key factors affecting the release of VOCs from materials. However, existing research on plastic track VOCs has mainly focused on detection methods and composition analysis. So, in this research, a PU plastic track was used as a case study. A 0.1 m^3^ environmental chamber was employed to simulate the release of VOCs under real environment conditions with different temperatures, RH and AER; air sampling canisters were used to collect the released VOCs. A qualitative and quantitative analysis of released VOCs was conducted by gas chromatography-mass spectrometry (GC-MS). This research aims to determine the types, principal components and mass concentration of VOCs and to discuss the impact of environmental factors including temperature, RH and AER on the VOCs released from a plastic track, providing evidence to avoid the effects of VOCs released from a plastic track on human health.

## 2. Materials, Instruments and Methods

### 2.1. Materials

A PU plastic track has a double-layer structure, which we prepared as follows. The lower layer was composed of PU adhesive and rubber particles (particle size: 2.00–3.00 mm, rubber content: 15%) uniformly mixed with a mass ratio of 3:1, and its thickness was (10.00 ± 1.00) mm. The upper layer was composed of PU adhesive and rubber particles (particle size: 1.00–2.00 mm, rubber content: 20%) uniformly mixed with a mass ratio of 1:1, and 40.00 g diluent was added into each 2.00 kg PU adhesive. The thickness of the upper layer was 3.00 ± 1.00 mm. The prepared plastic track was displayed outdoors for 20 d, with the outdoor environment free of VOC pollution around the plastic track and, once fully formed, it was sealed in a polytetrafluoroethylene film and stored at 25 ± 5 °C.

The sample was taken out 24 h before the experiment, and a square of 200.00 mm × 200.00 mm was cut 50.00 mm from the periphery of the sample. The cut sample was sealed and covered with aluminum foil in the side and bottom parts, ensuring an exposed area of 0.04 m^2^, as seen in Figure 1. Then, the sample was pre-equilibrated for (24 ± 1) h in a pure environment of 23 ± 2 °C and 50 ± 10% RH, and subsequently, the PU plastic track sample released VOCs in the environmental chamber.

### 2.2. Experiment Instruments and Reagents

A VOC release environmental chamber (0.1 m^3^, Shanghai Qinpei, Shanghai, China), QP 21-H4L100, was used for experimental tests, as seen in Figure 2. The following ancillary equipment complements the experimental installation: Agilent 7820A/5977E GC-MS (Agilent, Santa Clara, CA, USA); Gilian IAQ-Pro constant flow air sampling pump (Sensidyne, St. Petersburg, FL, USA); Silonite SUMMA canisters (Entech, California, USA); 7100A preconcentrator (Entech, Simi Valley, CA, USA); 7016CA autosampler (Entech, California, USA); 3100A canister cleaner (Entech, Simi Valley, CA, USA); and 4600A automatic VOCs dilutor (Entech, Simi Valley, CA, USA).

The internal standard used was (all from Linde, Munich, Germany) a four-component gas mixture (CH_2_BrCl, C_6_ClD_5_, C_6_H_4_F_2_, C_6_H_4_BrF, 4 ppb); a PAMS mixed standard gas (C_2_-C_12_, 57 types, 4 ppb); and a US EPA TO-15 mixed standard gas (65 types, 4 ppb).

Moreover (all from HUSHI, Chongqing, China), deionized water (0.1 μS·cm^−1^), alkaline cleaning agent (pH ≥ 7.5), nitrogen and helium (purity ≥ 99.999%) were employed.

### 2.3. Methodology

The mass concentrations of VOCs in the cleaned environmental chamber and canisters were ≤50.00 μg·m^−3^, and the mass concentrations of other single pollutants were ≤5.00 μg·m^−3^ [27]. Meanwhile, the Silonite canisters were vacuumed to ≤6.66 Pa for standby [28].

After silylation treatment and cleaning, the GC-MS instrument was used for the blank test. The mass concentrations of the targeted compounds were lower than the detection limit.

The temperature, RH, AER and other parameters of the chamber were set. After no-load operation until the parameters in the chamber reached the preset values, and after stabilization for 1 h, the plastic track sample was put in the center of the chamber (the specific parts as seen in Figure 3). The chamber door was then closed for the VOC release test. The instant when the sample was put into the chamber was recorded as initial time 0.

After the release of VOCs from the plastic track for 24 h, the canisters were employed for the constant current collection of VOCs. The sampling flow was 0.20 L·min^−1^, and the sampling time was 30 min.

For the qualitative and quantitative analysis of VOCs, the cold trap concentration conditions were the following: for the glass-bead cold trap concentration, the flow rate was 100.00 mL·min^−1^ at −150 °C; the resolution temperature was 10 °C; the valve temperature was 100 °C; and bakeout was performed at 150 °C for 15 min. For the Tenax tube cold trap concentration, the flow rate was 10.00 mL·min^−1^ at −15 °C; the resolution temperature was 180 °C for 3.5 min; and bakeout was carried out at 190 °C for 15 min. For the capillary glass tube absorption, focusing was performed at −160 °C; the resolution time was 2.5 min; and bakeout was carried out at 200 °C for 5 min. Finally, the transmission line temperature was 120 °C.

As far as the double column chromatographic conditions are concerned, a PLOT capillary column (15 m × 0.32 mm × 3 μm) and a DB-624 capillary column (60 m × 0.25 mm × 1.4 μm) were used. The temperature program was as follows: initial temperature of 35 °C, kept for 3 min; temperature rise to 180 °C at 6 °C·min^−1^, kept for 7 min; temperature rise to 200 °C at 10 °C·min^−1^, kept for 4 min. The sample inlet temperature was 200 °C. The PLOT and DB-624 column carrier gas flow rates were 1.00 and 1.30 mL·min^−1^, respectively. The temperature of the FID detector was 200 °C, and the solvent delay time was 5.6 min.

As far as the mass spectrometry conditions are concerned, the interface temperature was 250 °C, the electron impact ionization was 70 eV, the MS quadrupole temperature was 150 °C, the auxiliary heating temperature was 200 °C, the ion source temperature was 230 °C, and SIM scanning was performed at 35–300 amu.

### 2.4. Experimental Conditions for VOC Release Tests

Exposed to outdoor conditions for a long time, the surface temperature of a plastic track can reach values as high as 60 °C [29,30]. Therefore, combined with the pre-equilibrated temperature of 23 ± 2 °C, the environmental temperature of VOC emission was determined in a range of 23–60 °C. The RH range was adjusted according to the actual RH under different temperatures, from 5 to 65%. The AER was set at 0.5–1.5 h^−1^. Table 1 shows the experimental conditions adopted for tests A to F, with the values of the three parameters commented on above.

## 3. Results

### 3.1. VOCs Release at Varying Temperature

Due to the large variation of RH under actual conditions, in this study we investigated low-temperature (≤40 °C; Figure 4) and high-temperature (> 40 °C; Figure 5) conditions to ensure consistent RH values. Table 2 complements the analysis of the results.

With the increase of temperature, the concentrations of VOCs gradually increase. Indicating with *C* the mass concentration, at 23 °C and 30 °C, it can be shown that *C*_alkane_ > *C*_halogenated hydrocarbon_ > *C*_oxygenated organic compounds_ > *C*_aromatic hydrocarbons_ > *C*_alkene_ > *C*_nitrogenous organic compounds_ > *C*_alkyne_. At 30 °C, *C*_alkane_ showed a rapid increase, and when the temperature was higher than 35 °C, *C*_oxygenated organic compounds_ significantly rose to its maximum value. Under the temperature range of 55–60 °C, *C*_halogenated hydrocarbon_ showed relatively rapid growth. A significant increase in the concentrations of some main VOC (n-butane, 3-methylheptane, n-octane, trans-2-pentene, ethylbenzene, 1,2-dichloroethene and acetone) as the temperature increased was noticed. On the other hand, some VOCs, including n-decane, toluene, styrene, dichloromethane, 1,1,2,2-tetrachloroethane, hexanal and acetonitrile, had a non-monotonic concentration trend (increasing and then slightly decreasing to equilibrium). *C*_n-octane_, *C*_n-decane_ and *C*_3-methylheptane_ showed larger values at 23 °C and 30 °C, while *C*_acetone_ rapidly increased to its maximum value at more than 35 °C, followed by *C*_n-octane_, *C*_1,2-dichloroethene_, *C*_n-decane_, *C*_3-methylheptane_, etc. In addition, dodecane was not detected under all temperatures, and acetylene, 1,1-dichloroethane and 2,3-dimethylbutane were not detected or detected at extremely low values to some extent.

### 3.2. VOCs Release at Varying Relative Humidity

Figure 6 and Figure 7 show the concentrations of different types of VOCs released from the plastic track under an RH of 45%, 55% and 65% at 30 °C and an RH of 5%, 10% and 20% at 60 °C, respectively. Table 3 gives more detail on the results.

An increase in VOC concentration at a given temperature was observed when the RH was increased. *C*_alkane_ was significantly higher than the concentrations of the other six organic compounds under different RH at 30 °C. However, *C*_oxygenated organic compounds_, *C*_alkane_ and *C*_halogenated hydrocarbon_ were significantly higher than the concentrations of the other four organic compounds at 60 °C.

Only *C*_n-decane_ (30 °C, 60 °C) and *C*_ethylbenzene_ (30 °C) decreased, while the concentrations of the other VOCs increased at a given temperature and increasing humidity. *C*_n-octane_, *C*_n-decane_ and *C*_3-methylheptane_ were higher at 30 °C, and *C*_acetone_ was the highest, followed by *C*_1,2-dichloroethene_, *C*_n-octane_, *C*_3-methylheptane_ and *C*_n-decane_ at 60 °C.

Additionally, dodecane was not detected, and the concentrations of methyliodide, 1,1-dichloroethylene, trichloroethylene, 1,1,1-trichloroethane, 1,2-dibromoethane were merely detected in a range of 0.02–0.05 μg·m^−3^.

### 3.3. VOCs Release at Varying Air Exchange Rate

Figure 8 and Figure 9 illustrate results under an AER of 0.5 h^−1^, 1.0 h^−1^ and 1.5 h^−1^ at 30 °C and 60 °C, respectively, complemented by data listed in Table 4.

The VOC concentrations significantly decreased with the increase of AER. Under different AERs, *C*_alkane_ was the highest, followed by *C*_halogenated hydrocarbon_, *C*_oxygenated organic compounds_ and *C*_aromatic hydrocarbons_ at 30 °C, while *C*_oxygenated organic compounds_ was the highest, followed by *C*_alkane_ and *C*_halogenated hydrocarbon_, at 60 °C.

Under different AER values, *C*_n-octane_, *C*_n-decane_, *C*_3-methylheptane_, *C*_hexanal_ and *ρ*_1,1,2,2-tetrachloroethane_ were higher at 30 °C, while *C*_acetone_ was the highest, followed by *C*_1,2-dichloroethene_, *C*_n-octane_, *C*_3-methylheptane_ and *C*_n-decane_ at 60 °C. In addition, only dodecane was undetected under AER = 0.5 h^−1^ and 1.0 h^−1^. Under AER = 1.5 h^−1^ at 30 °C, twelve VOC compounds were not detected, namely 2,3-dimethylbutane, dodecane, trans-2-butene, 1,3-butadiene, isobutene, bromomethane, 1,1-dichloroethane, 1,1,1-trichloroethane, 1,2-dibromoethane, methyliodide, 1,1-dichloroethylene and trichloroethylene; at 60 °C, there were seven non-detected species, namely 2-methylhexane, dodecane, isobutene, trichlorotrifluoroethane, 1,1-dichloroethane, 1,1,1-trichloroethane and methyliodide.

## 4. Discussion

### 4.1. Role of Temperature

The ambient temperature can change the thermal motion and vapor pressure of VOC molecules, thus affecting the adsorption ability and absorption capacity of materials for VOC molecules, leading to the variation in the mass concentrations of VOCs. As shown in Figure 4 and Figure 5, environmental temperature can steadily promote the emission of VOCs from a plastic track.

Regarding the molecular thermal motion, An et al. [16] thought that with the increase of temperature, the emission of VOCs is accelerated, which is related to the molecular thermal motion of VOCs. In our study, the increase of temperature enhanced the molecular thermal motion of VOCs inside the plastic track, leading to a lower adsorption ability and absorption capacity for the VOC molecules. This triggered the release of VOC molecules in quantity, thus causing an increase in the concentrations of VOCs. Regarding the vapor pressure of VOC molecules, Verheyen et al. [31] found that the composition and emission of VOCs change with the increase in temperature, which is related to the vapor pressure of VOC molecules. According to the Antoine equation, it is
(1)$$\begin{array}{c} lg\;P=A-\frac{B}{T+C} \end{array}$$
where *A*, *B* and *C* are the physical property constants varying with the compound and are greater than 0; *P* is the vapor pressure of the compound atm; and *T* is the absolute temperature in K. The Antoine Equation (1) indicates that the vapor pressure of VOC molecules inside the plastic track increased with the increase of temperature, promoting the release of VOCs from the plastic track to the air, thus leading to the increase in the concentrations of VOCs.

Moreover, the emission of VOCs is significantly correlated with their diffusion coefficient, which obviously increases with temperature, as also discussed by Yang et al. [32] and Zhang et al. [33]. Wei et al. [18] indicated an exponential relationship between the diffusion coefficient of VOCs and the environmental temperature. Yang et al. [34] simulated the emission of VOCs under different environmental temperatures and found that the relationship between the diffusion coefficient of VOCs and the temperature follows the Arrhenius equation:(2)$$\begin{array}{c} K=Ae^{-\frac{E_{a}}{RT}} \end{array}$$
where *K* represents the parameter (in our case, the diffusion coefficient) at a certain temperature; *R* is the molar gas constant, J·mol^−1^·K^−1^; *E_a_* is the diffusion activation energy, regarded as a constant independent of *T*, J·mol^−1^; and *A* is the pre-exponential factor, with the same unit as *K*.

This study shows that the increase of temperature leads to an increase in the thermal motion of VOCs, causing a noticeable accelerated diffusion rate and leading to an increased mass concentration of VOCs.

In addition, Figure 4 and Figure 5 indicate an enhanced activity of VOC emission from the plastic track at temperatures of 30–35 °C and > 50 °C. Previous research found that, under different temperature ranges, the activity of VOC emissions from materials is not relevant, and that the concentrations of VOCs fluctuate. Huang et al. [35] studied the effect of temperature on the release of VOCs from furniture wood finishes, and showed that the effect of different temperatures on the emission of VOCs varied, with a more pronounced increase in VOC concentration between 15 °C and 25 °C and a relatively small difference in VOCs peak concentration between 25 °C and 35 °C. Jo et al. [36] analyzed the effect of temperature on the release of indoor VOCs, and showed that VOC concentrations increased significantly at temperatures greater than 25 °C, and the regression curves of the total VOCs appeared to increase as temperature rises. The results of this study for plastic tracks are in strong agreement with the above results, while the differences in the active temperature range of VOCs may be due to differences in materials. Thus, future research on plastic tracks can be focused on the tendency and mechanism of the variation in VOC concentrations under temperatures of 30–35 °C and > 50 °C.

Furthermore, it is worth noting that, under different environmental temperatures, the detection rate of VOC species is 98.04–99.02%. Alkane, oxygenated organic compounds, halogenated hydrocarbon and aromatic hydrocarbons are the main types of VOCs, which corresponds to the conclusion proposed by Chang et al. [1] that alkane, halogenated hydrocarbon and benzenes are commonly detected in VOCs released from plastic tracks. Our study indicates that n-butane, 3-methylheptane, n-octane and n-decane predominate in the class of released alkanes; trans-2-pentene predominates among alkenes; toluene, ethylbenzene and styrene are the most relevant in aromatic hydrocarbons; hexanal and acetone in oxygenated organic compounds; and acetonitrile predominates in nitrogenous organic compounds. This finding is consistent with the conclusion by Cui et al. [5] and Guo et al. [8] that toluene, ethylbenzene, xylene, 1,2-dichloroethene, dichloromethane, 1,2-dichloropropane, styrene and 1,3,5-trimethylbenzene are the most relevant detected VOCs released from plastic tracks. At 30 °C, the accelerated release of n-octane, n-decane and 3-methylheptane leads to a rapid growth in *C*_alkane_. At 35 °C, the large release of acetone causes a significant increase in *C*_oxygenated organic compounds_. At 55–60 °C, the rapid release of 1,2-dichloroethene results in the continuous growth of *C*_halogenated hydrocarbon_. The rapid release of the aforementioned VOC species may be closely related to the vapor pressure of different VOCs at varying temperatures. According to the Antoine equation, the vapor pressure of different compounds is closely related to the physical constants A, B and C in Equation (1). Yaws et al. [37] gave the values for A, B and C for different compounds at different temperature ranges, which can be used to calculate their vapor pressure; the differences in vapor pressure will directly lead to differences in the release of different compounds at changing temperature.

In addition, a relationship between the main types of released VOCs and the raw material can be highlighted. Kamarulzaman et al. [7] detected that benzenes, aldehydes and ketones are the predominant VOCs released from rubber particles. Jiang [6] indicated that PU adhesive can release hydrocarbons that include alkane, cycloalkane and aromatic hydrocarbons with their derivatives, as well as aldehydes and ketones. Li et al. [10] indicated that PU adhesive can release nine VOCs, including benzenes, butyl acetate and undecane. Zhou et al. [9] observed that diluent leads to the formation of halogenated hydrocarbons and organic matter including aldehydes, ketones and nitriles. Moreover, the combination of a PU adhesive and a diluent can release 16 types of VOCs, including dichloromethane, trichloromethane, and carbontetrachloride. The VOCs released from raw materials above are consistent with VOCs released from plastic tracks.

The impact of plastic track VOCs on human health can be seen from the main classes and species of VOCs released from plastic track and their raw materials. Snyder et al. [38] showed that aromatic hydrocarbons such as benzene, toluene, and ethylbenzene in ambient air not only irritate human skin and mucous membranes, but also cause chronic and acute damage to the human respiratory, hematopoietic, and neurological systems. Axelsson et al. [39] stated that benzene concentration was significantly and positively associated with the incidence of asthma. Tong et al. [40] and DE-Miranda et al. [41] showed that halogenated hydrocarbons such as dichloromethane and 1,1,2,2-tetrachloroethane in ambient air can enter the human body directly through breathing and damage people’s cardiovascular system and respiratory system. Zhang et al. [42] and Yang et al. [43] pointed out that acetone has irritating effects on people’s eyes, nasal mucosa, and nerve endings, and vomiting, shortness of breath, cramps, and even coma can occur in acute poisoning. As the main VOCs released from plastic track and their raw materials, the above-mentioned VOCs are more consistent with the discomfort reactions that occur during people’s exercise on plastic tracks. Meanwhile, in addition to the main VOCs released from the plastic track mentioned above, the combined reaction of other VOC monomers can pose a greater threat to people’s health.

### 4.2. Role of Relative Humidity

The variation in environmental RH causes a change in environmental water vapor pressure and in the water vapor pressure gradient inside the material, as well as in the adsorption characteristics of the VOCs, leading to a constant change in their concentration. As shown in Figure 6 and Figure 7, the increase in RH can promote the emission of VOCs from plastic track.

Previous research agrees that the release of VOCs is proportional to the RH [21,44]. In the emission of VOCs from plastic tracks, the increase of relative humidity causes an increase in the vapor pressure, and the diffusion coefficient of most VOCs not reacting with water vapor increases as well, thus promoting their release. In addition, the increase of the environmental vapor pressure reduces the vapor pressure gradient between the external environment and the interior of the plastic track, and the slow evaporation of water inside the plastic track reduces the resistance to VOC release. Therefore, a higher RH has less impact on the release of VOCs, which is a beneficial aspect.

In any case, the release of VOCs from materials is closely related to their porous properties and adsorption characteristics [45]. As a typical porous material, a plastic track can adsorb hydrophilic and hydrophobic molecules. These molecules occupy a certain pore space, and the total pore space is constant. With the increase of RH, the water evaporation becomes slower, more water molecules are absorbed by the hydrophilic units, and the occupied pore space gradually becomes larger. Thus, hydrophobic molecules such as alkanes and halogenated hydrocarbons in VOCs occupy less pore space and are released from the plastic track. At the same time, hydrophilic molecules such as oxygenated organic compounds in VOCs are squeezed by water molecules to accelerate their release.

Figure 6 and Figure 7 also indicate that, although environmental RH promotes the emission of VOCs from plastic tracks, the increase in the concentration of VOCs is significantly smaller than that caused by the increase of temperature, which is consistent with the research of Lin et al. [20] and Zheng et al. [46] on plywood, showing that the role of temperature is more relevant than the role of humidity.

Furthermore, it is worth noting that the change of RH shows no effect on the percentage of the concentrations of different VOCs released from plastic tracks in total VOC, and that the main classes and species of VOCs are still basically consistent with those detected at changing temperatures. The detection rate of VOC species is 98.04–99.02%. However, under different RH conditions and referring to different temperatures, the concentrations of different classes and species of VOCs show a small variation, further indicating that RH has a lower effect on the release of VOCs from the plastic track than the temperature.

### 4.3. Role of Air Exchange Rate

The change of AER alters the concentration gradient of VOCs in the boundary layer between the material and the air flow in the chamber, and also changes the VOC’s diffusion coefficient and adsorption effect, eventually leading to the change in VOC concentrations. Figure 8 and Figure 9 show that the increase of AER reduces the concentrations of VOCs released from a plastic track.

In previous research, Yang et al. [47], Deng et al. [24] and Manoukian et al. [26] indicated that the increase of AER promotes the early emission rate of VOCs. Chang et al. [48] and Xiong et al. [49] pointed out that the emission of VOCs from materials is controlled by the concentration gradient of VOCs in the boundary layer between the material and the air flow, following Fick’s second law:(3)$$\begin{array}{c} \frac{\partial C}{\partial t}=D\frac{\partial^{2}C}{\partial x^{2}} \end{array}$$
where *C* is, in Equation (3), the volume concentration of the diffusive substance; *t* is the diffusion time; *x* indicates the diffusion direction; and *D* is the diffusion coefficient of the diffusive substance inside the material that can be substantially considered as constant and it is expressed as [length^2^/time].

When the boundary layer concentration gradient $\frac{\partial C}{\partial x}$ increases, $\frac{\partial C}{\partial t}$ increases as well, leading to a release of VOC molecules inside the plastic track from the zones at high concentration to those at low concentration, thus causing a rapid increase in the concentration of VOCs.

As a matter of fact, our research shows that the increase of AER promotes the early release of VOCs from materials. However, the emission time of VOCs in current research is 24 h. Yang et al. [4] and Gan et al. [50] pointed out that VOCs reach an equilibrium in 12–16 h in the environmental chamber after their release from the plastic track. Therefore, the collection time of the VOCs in this study had reached the late stage of their release. At this moment, the increase of AER led to an increase in the volume of pure air entering the chamber, taking away more VOCs than the VOC’s accelerated release by the concentration gradient change and leading to a rapid decline in the concentrations of VOCs. Simultaneously, the concentration gradient of the boundary layer decreased, weakening the release of VOCs. Therefore, in the late stage of VOC emission, the AER was negatively correlated with the concentration of VOCs, which is consistent with the studies of Héroux et al. [51], Liu et al. [52] and Yang et al. [53] on the long-term emission of VOCs from materials. In future research, a real-time detection of VOCs on plastic tracks should be carried out to determine the impact of AER on the early release of VOCs from plastic tracks.

In addition, Figure 8 and Figure 9 and Table 4 show that, although the increase of AER reduces the concentrations of different classes and species of VOCs, the increase of environmental temperature can significantly increase the concentrations of VOCs under different AER, indicating that temperature has a more significant impact on the concentrations of VOCs than AER does. This is consistent with the conclusion of Lin et al. [20] that environmental temperature has the greatest impact on the emission of VOCs from materials. Moreover, the variation in the concentrations of VOCs caused by the change of AER is more significant than what we have observed in terms of relative humidity variation.

It is worth noting that the detection rate of VOC species declines with a significant increase in AER. Dodecane is not detected under all environmental conditions, and some VOCs with low concentrations are not detected when AER increases. This indicates that when the AER increases, the volume of pure air entering the chamber per unit time increases, taking away some VOC species with low concentrations and leading to their non-detection or to a further decline in the concentration of VOC compounds.

## 5. Conclusions

In this work, the release of VOCs from the plastic track was carried out using a 0.1m^3^ environmental chamber at a temperature range of 23–60 °C, with a relative humidity of 5–65% and an AER of 0.5–1.5 h^−1^; the qualitative and quantitative analysis of the VOCs was carried out by GC-MS. The main findings of this study are as follows:The increase of environmental temperature and relative humidity can promote the release of VOCs from plastic tracks, while the increase in air exchange rate can significantly reduce this effect.Environmental temperature, relative humidity and air exchange rate have no definite impact on the main classes and species of VOCs released from plastic tracks—the main classes include alkane, alkene, aromatic hydrocarbons, halogenated hydrocarbon, oxygenated organic compounds and nitrogenous organic compounds, and the main species include n-butane, 3-methylheptane, n-octane, n-decane, trans-2-pentene, toluene, ethylbenzene, styrene, dichloromethane, 1,1,2,2-tetrachloroethane, 1,2-dichloroethene, hexanal, acetone, acetonitrile. However, the increase of the environmental temperature can significantly change the percentage of the mass concentrations of different VOCs in total VOC; when the temperature was less than or equal to 30 ℃, *C*_alkane_ > *C*_halogenated hydrocarbon_ > *C*_oxygenated organic compounds_ > *C*_aromatic hydrocarbons_ > *C*_alkene_ > *C*_nitrogenous organic compounds_ > *C*_alkyne_, and when the temperature was higher than 35 °C, *C*_oxygenated organic compounds_ significantly rose to its maximum value because *C*_acetone_ rapidly increased.Environmental temperature and relative humidity cannot significantly exert an impact on the detection rate of VOC species released from a plastic track, while air exchange rate is the main factor leading to the non-detection of VOCs species; when the air exchange rate was increased from 0.5 h^−1^ to 1.5 h^−1^, the number of non-detected VOC species increased from 1 to 7–12.Environmental temperature has the most significant effect on the emission of VOCs from plastic tracks, followed by air exchange rate and relative humidity.

## Figures and Tables

**Figure 1 ijerph-20-01828-f001:**
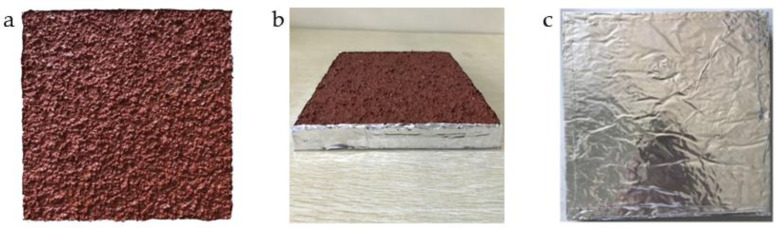
A PU plastic track sample after sealing and covering with aluminum foil. (**a**) Front, (**b**) side, and (**c**) back surfaces.

**Figure 2 ijerph-20-01828-f002:**
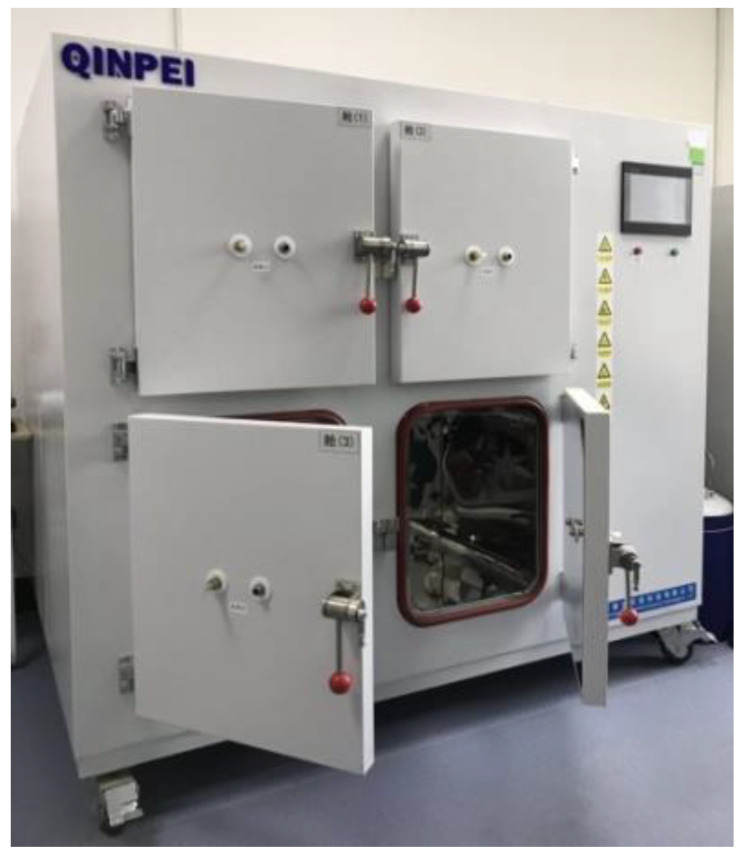
Physical picture of environmental chamber.

**Figure 3 ijerph-20-01828-f003:**
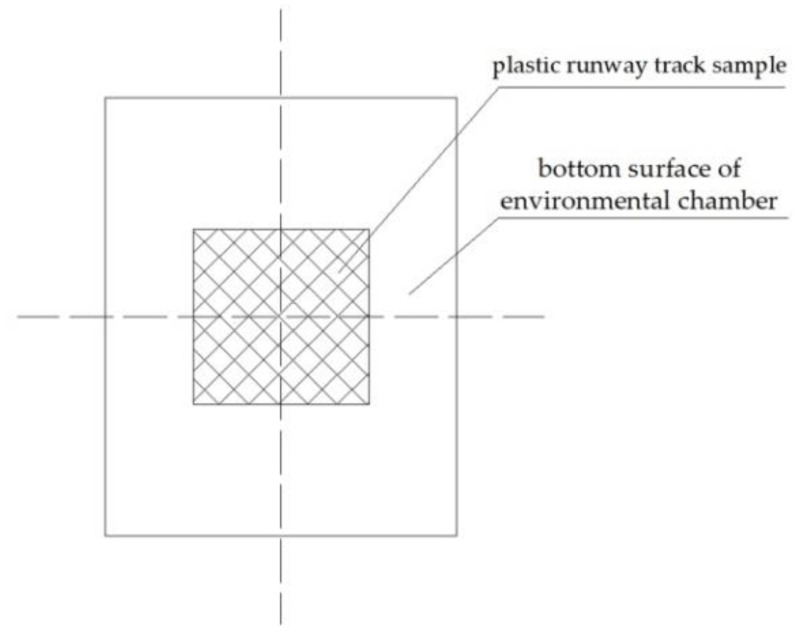
Specific parts of the plastic track sample placed in the environmental chamber.

**Figure 4 ijerph-20-01828-f004:**
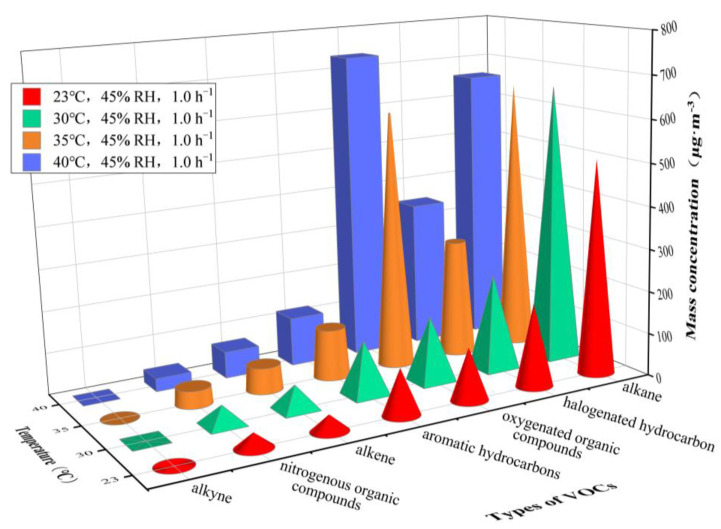
Mass concentrations of VOCs released from plastic track at temperatures varying from 23 to 40 °C. RH = 45% and AER = 1.0 h^−1^.

**Figure 5 ijerph-20-01828-f005:**
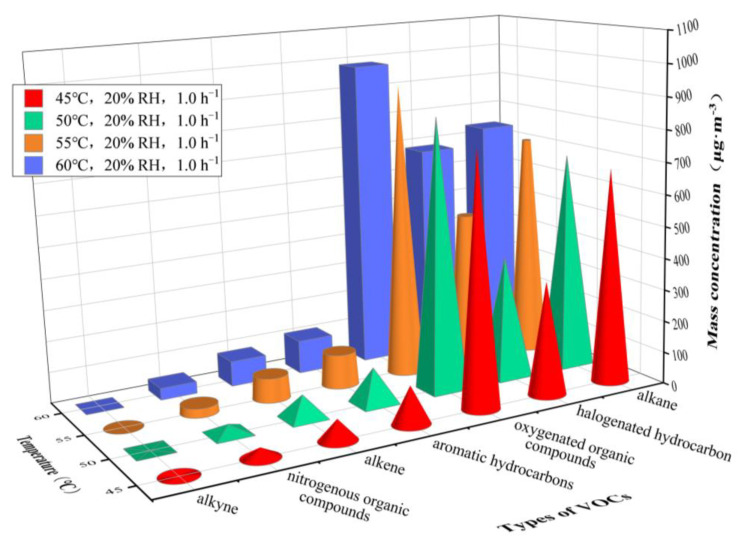
Mass concentrations of VOCs released from plastic track at temperatures varying from 45 to 60 °C. RH = 20% and AER = 1.0 h^−1^.

**Figure 6 ijerph-20-01828-f006:**
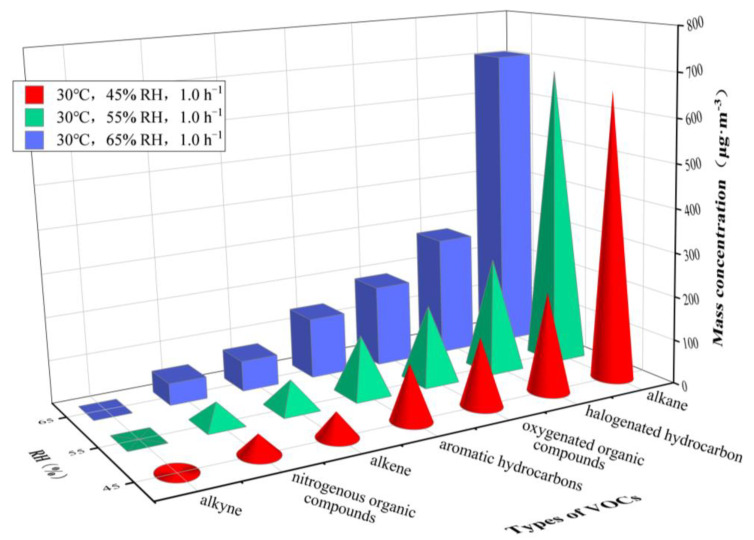
Mass concentrations of VOCs released from plastic track at RHs varying from 45 to 65%. Temperature = 30 °C and AER = 1.0 h^−1^.

**Figure 7 ijerph-20-01828-f007:**
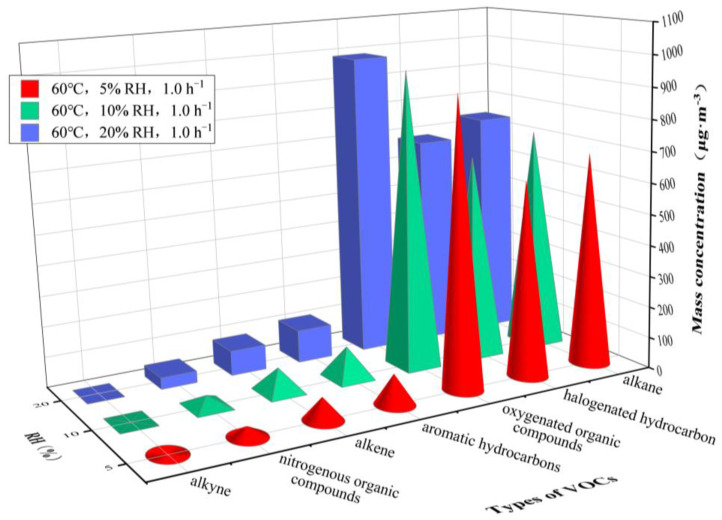
Mass concentrations of VOCs released from plastic track at RHs varying from 5 to 20%. Temperature = 60 °C and AER = 1.0 h^−1^.

**Figure 8 ijerph-20-01828-f008:**
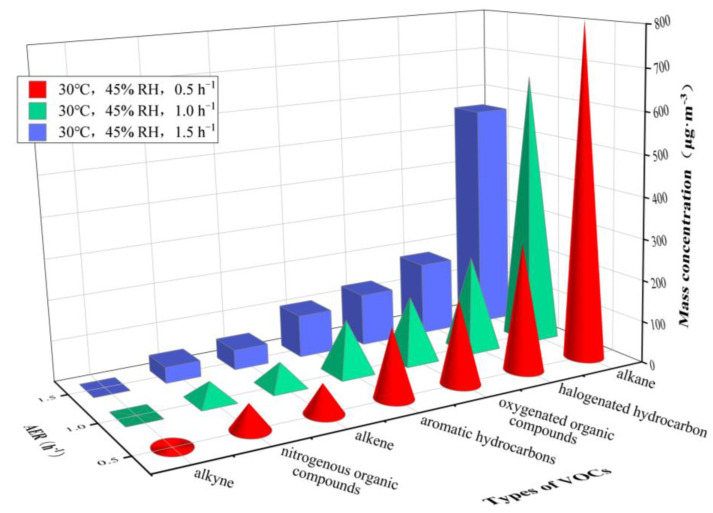
Mass concentrations of VOCs released from plastic track at AERs varying from 0.5 to 1.5 h^−1^. Temperature = 30 °C and RH = 45%.

**Figure 9 ijerph-20-01828-f009:**
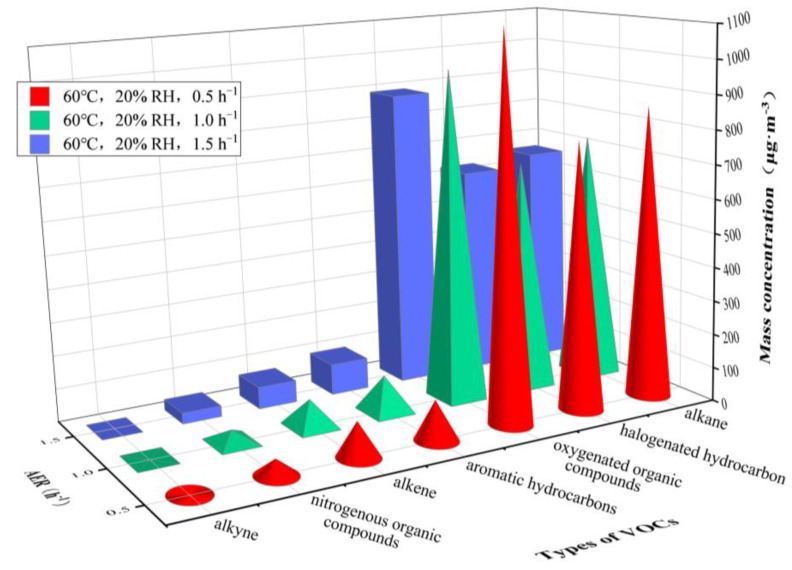
Mass concentrations of VOCs released from plastic track at AERs varying from 0.5 to 1.5 h^−1^. Temperature = 60 °C and RH = 20%.

**Table 1 ijerph-20-01828-t001:** Experimental parameters.

Test Group	Temperature/°C	RH/%	AER/h^−1^
A	23/30/35/40	45	1.0
B	45/50/55/60	20	1.0
C	30	45/55/65	1.0
D	60	5/10/20	1.0
E	30	45	0.5/1.0/1.5
F	60	20	0.5/1.0/1.5

**Table 2 ijerph-20-01828-t002:** Mass concentrations of main VOCs released from plastic track under different environmental temperatures.

Types of VOCs	Main VOCs Species	Mass Concentrations/μg·m^−3^							
		45% RH, AER = 1.0 h^−1^				20% RH, AER = 1.0 h^−1^			
		23 °C	30 °C	35 °C	40 °C	45 °C	50 °C	55 °C	60 °C
alkane	n-butane	40.71	45.94	48.84	50.15	59.37	54.70	55.21	56.57
	3-methylheptane	106.40	139.02	141.35	146.00	155.81	168.49	180.70	198.62
	n-octane	159.17	209.06	212.90	226.40	227.74	237.37	245.69	255.90
	n-decane	138.80	197.84	179.36	165.96	164.72	157.37	159.49	161.48
alkene	trans-2-pentene	28.24	47.06	50.69	52.96	54.59	61.42	65.56	70.02
aromatic hydrocarbons	toluene	17.89	22.82	21.28	20.71	22.04	18.90	16.61	14.81
	ethylbenzene	9.84	12.84	10.74	9.45	10.15	10.85	11.12	12.50
	styrene	49.19	60.65	60.56	59.78	61.84	58.44	55.59	54.34
halogenated hydrocarbon	dichloromethane	10.32	14.97	14.41	13.47	13.13	12.96	11.72	9.89
	1,1,2,2-tetrachloroethane	65.16	76.65	82.42	76.87	75.85	73.72	69.15	64.35
	1,2-dichloroethene	64.76	82.69	136.17	216.19	225.29	260.75	365.46	569.33
oxygenated organic compounds	hexanal	72.41	93.44	90.18	87.85	84.69	79.90	76.15	73.54
	acetone	30.27	35.58	492.20	615.53	677.18	744.46	819.88	872.14
nitrogenous organic compounds	acetonitrile	34.97	45.02	40.98	31.34	31.30	29.90	28.58	37.91

**Table 3 ijerph-20-01828-t003:** Mass concentrations of main VOCs released from plastic track under different values of relative humidity.

Types of VOCs	Main VOC Species	Mass Concentrations/μg·m^−3^					
		30 °C, AER = 1.0 h^−1^			60 °C, AER = 1.0 h^−1^		
		45%	55%	65%	5%	10%	20%
alkane	n-butane	45.94	54.92	62.45	40.37	49.83	56.57
	3-methylheptane	139.02	153.97	169.30	185.52	192.09	198.62
	n-octane	209.06	214.91	220.92	235.93	248.17	255.90
	n-decane	197.84	189.79	183.92	169.26	165.18	161.48
alkene	trans-2-pentene	47.06	51.40	55.17	63.14	67.35	70.02
aromatic hydrocarbons	toluene	22.82	25.55	27.71	13.38	14.29	14.81
	ethylbenzene	12.84	12.20	11.88	11.86	12.28	12.50
	styrene	60.65	64.47	66.17	49.79	51.48	54.34
halogenated hydrocarbon	dichloromethane	14.97	16.13	17.76	8.56	9.40	9.89
	1,1,2,2-tetrachloroethane	76.65	81.91	83.60	59.73	62.90	64.35
	1,2-dichloroethene	82.69	102.02	114.52	528.26	551.09	569.33
oxygenated organic compounds	hexanal	93.44	98.06	103.32	70.53	72.18	73.54
	acetone	35.58	37.21	40.59	823.40	855.35	872.14
nitrogenous organic compounds	acetonitrile	45.02	47.94	50.13	33.28	35.96	37.91

**Table 4 ijerph-20-01828-t004:** Mass concentrations of main VOCs released from plastic track under different values of air exchange rate.

Types of VOCs	Main VOC Species	Mass Concentrations/μg·m^−3^					
		30 °C, 45% RH			60 °C, 20% RH		
		0.5 h^−1^	1.0 h^−1^	1.5 h^−1^	0.5 h^−1^	1.0 h^−1^	1.5 h^−1^
alkane	n-butane	58.85	45.94	35.22	65.10	56.57	51.85
	3-methylheptane	173.96	139.02	107.49	239.14	198.62	176.66
	n-octane	251.20	209.06	187.14	293.16	255.90	235.70
	n-decane	226.15	197.84	181.31	181.01	161.48	147.42
alkene	trans-2-pentene	56.17	47.06	42.22	89.13	70.02	55.46
aromatic hydrocarbons	toluene	29.02	22.82	18.09	16.13	14.81	12.23
	ethylbenzene	16.31	12.84	10.05	14.01	12.50	11.40
	styrene	69.27	60.65	55.96	59.28	54.34	47.56
halogenated hydrocarbon	dichloromethane	19.78	14.97	10.82	12.27	9.89	9.45
	1,1,2,2-tetrachloroethane	101.77	76.65	64.19	72.44	64.35	61.78
	1,2-dichloroethene	90.98	82.69	73.89	656.14	569.33	526.86
oxygenated organic compounds	hexanal	113.86	93.44	82.75	79.73	73.54	70.14
	acetone	46.76	35.58	28.31	995.69	872.14	789.03
nitrogenous organic compounds	acetonitrile	60.83	45.02	39.21	48.48	37.91	32.68

## Data Availability

The datasets generated during and/or analyzed during the current study are available from the corresponding author on reasonable request.

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
