# Peer review of "Research on the Effects of Environmental Factors on the Emission of Volatile Organic Compounds from Plastic Track"

_ijerph, 2023, doi:10.3390/ijerph20031828_

Round 1

Reviewer 1 Report

Plastic runway tracks, which are composed of multiple organic hydrocarbons and their derivatives, are susceptible to the release of VOCs under the influence of external environmental factors such as temperature, relative humidity and air exchange rate, thus affecting the health of sports people. This study is aimed at this human health hazard and uses environmental chamber-GC-MS to carry out the release, collection and testing of VOCs released from plastic runways at different temperatures, relative humidity and air exchange rates, thus clarifying the influence of environmental factors on the release of VOCs from plastic runways. So, the research is of significant value and innovative.

In response to the full text, the following recommendations are made:

1. In the abstract section, the main experimental results and conclusions are given, but when the experimental results are given, the main quantitative results are lacking and important quantitative results need to be added here.

2. In the introduction, the authors provide an overview of the effects of external environmental factors such as temperature, relative humidity and air exchange rate on the release of VOCs, but the review is too extensive and needs to be further streamlined.

3. In the method section, environmental chambers are used for the release of VOCs from plastic runways. A photograph of the environmental chamber used should be given here, with a diagram showing the specific part of the plastic runway sample placed in the chamber.

4. In the results analysis section, in addition to the explanation of the causes for each part of the results, there needs to be some discussion of the specific health effects of the main components of VOCs, corresponding to their human health hazards in the previous section.

5. In the conclusion section, quantitative conclusions are lacking. Please give some of the quantitative conclusions in relation to the specific quantitative results in the previous section.

Reviewer 2 Report

In the present manuscript entitled "Research on the Effects of Environmental Factors on the Emission of Volatile Organic Compounds from Plastic Track" authors tried to investigate and quantify the concentration of volatile organic compounds (VOCs) from the plastic track under different environmental conditions like change in temperature, reative humidity and air exchange rate. A systematic experimental analysis is made using custom made plastic track sample kept in a VOC observation chamber under controlled environmental conditions. The inferences are made with good pictorial representations and tables. Though the manuscript is scientifically sound enough to be accepted for publications, there are few typographical and gramatical erros and some minor corrections are needed before getting accepted. Some of the specific comments are listed below

1. Abstract: "Through this paper, we can provide a basis for effectively avoiding the impacts of VOCs from plastic track" I could not see any recommendations made in this manuscript to avoid impact of VOCs from plastic track.  

2. Introduction:  Good performance in terms of what? Plant's health?or track durability?or something else please mention.

3. Introduction: "suggested that baking of paints before use could accelerate the emission of pollutants"- "suggested" means backing of paint is encouraged before use. Is it so?

4. Introduction : "increasing initial emission rate and decay rate of paints" - do you mean "decay rate of VOCs from paints with increasing temperature"?

6. Introduction : "96 hours at 20 °C, 30 °C and 50 °C, showing that that"- remove repeated "that"

7. Introduction : "building materials, found that increases in" - should be "increase in"

8. Introduction : "A 0.1 m3" can be "A 0.1m^3"

9. Materials: "Is there any change of ambient pollutants/VOC deposition on the plastic track when it is kept outdoor for 20 days? if so how to differentiate is from the VOCs actually emittted from the track?

10: Page 13: "tempearture, K" can be "tempearture in K"
